# Agricultural Stakeholders’ Perceptions of Occupational Health and Safety in the Southeastern U.S. Coastal States

**DOI:** 10.3390/ijerph18126605

**Published:** 2021-06-19

**Authors:** Tracy Irani, Beatrice Fenelon Pierre, Tyler S. Nesbit

**Affiliations:** Department of Family, Youth, and Community Sciences, University of Florida, Gainesville, FL 32611, USA; bfenelon.pierre@ufl.edu (B.F.P.); tnesbit@ufl.edu (T.S.N.)

**Keywords:** agriculture, occupational health and safety, barriers, enablers, COVID-19, farmworkers

## Abstract

Agriculture remains a highly dangerous industry for occupational health and safety. This study sought to understand the perspective of agricultural professionals with respect to the current state of the industry, challenges, and opportunities relevant to occupational health and safety. Additional questions related to the COVID-19 pandemic emerged in the findings as well. Eleven industry professionals were interviewed, and the transcripts were qualitatively analyzed for emergent themes following a constant comparative method. Three themes emerged in our findings: a description of the current state of occupational health and safety in the agricultural industry, barriers to improving occupational health and safety, and enablers of occupational health and safety. Each theme contained subthemes. The description of the industry encompassed regulations, inherent danger, and attitudes and education. Barriers included education, health care access, logistics, discrimination and cultural competency, economic considerations, and the labor contracting system. Enablers included education, regulations, and health care and prevention. These findings are consistent with existing literature, revealing interconnected and overlapping challenges and opportunities. Further research is recommended with a broader sample of participants, especially farmworkers.

## 1. Introduction

Agriculture is one of the most dangerous industries globally for worker health and safety. Up to 170,000 agricultural workers around the world die at work each year, representing approximately half of all fatal occupational accidents [1]. In the U.S., agricultural workers involved in farming, forestry, fishery, and hunting endured 573 fatal work injuries in 2019, more than any sector except for construction and truck transportation [2]. Risks to occupational health and safety include adverse weather, intense labor, and exposure to mechanical, biological, and chemical dangers. U.S. agricultural workers suffer the highest rate of heat-related fatalities of any industry, with a rate of 3.06 deaths per million workers per year. This is nearly 3-fold higher than in construction, the industry with the second highest rate at 1.13 [3]. Symptoms ranging from headache and dizziness to dehydration and heat stroke are common examples of heat-related illness (HRI) associated with exposure to extreme heat and intense working conditions [4]. Evidence indicates that heat stress impacts the risk perceptions of exposed individuals so that they perceive behavior to be less risky and engage in more risky behaviors [5]. This phenomenon is particularly dangerous for agricultural workers, who may operate or work with heavy machinery and dangerous equipment, such as tractors and harvesting machines, posing an elevated threat to worker health and safety. Rashes and allergic reactions are often triggered by biological agents such as pollen, plants, and insects present in agricultural settings. Additionally, exposure to pesticides and other agricultural chemicals elevates risks of a host of diseases, including cancer, asthma, diabetes, Parkinson’s disease, and cognitive impairment [6]. Furthermore, recent research has highlighted the challenges to mental health that agricultural workers face due to a variety of occupation-related stressors [7].

In addition to the inherent dangers of modern production agriculture outlined above, the prevalent socioeconomic conditions of agricultural workers also impact occupational health and safety. The median annual wage of U.S. agricultural workers in May 2019 was $25,840 [8]. Most U.S. agricultural laborers are immigrants, and nearly half of crop farmworkers do not have legal immigration status. The H-2A program allows U.S. employers to hire foreign nationals in temporary agriculture jobs. For those working under the H-2A visa program, the southeastern states offer the lowest wages in the country, less than $12 per hour [9]. For workers sending money back home for their families, these wages place them in impoverished conditions. This may result in insecure housing arrangements, such as tenuous rental agreements and overcrowding [10]. These circumstances constitute an additional stressor on workers and their families as well as a direct threat to health and safety for proper sanitation, such as pesticide safety practices [11].

These socioeconomic conditions, combined with a piece-rate payment structure typical in agricultural operations, create an incentive to maximize output, contributing to behaviors such as working through break times prescribed for rest, shade, and hydration and working longer hours overall. This situation is exacerbated by laws that do not require agricultural workers to receive overtime pay for working beyond the standard 40-h work week. In recent years, several policy efforts have attempted to address the challenges of farmworker health and safety. The federal office responsible for agricultural worker safety is the Occupational Safety and Health Administration (OSHA), housed within the U.S. Department of Labor. The minimum standards set by OSHA are further addressed at the state level. In 2015, the U.S. Environmental Protection Agency (EPA) released the Agricultural Worker Protection Standard (WPS), which targets the level of pesticide exposure of agricultural workers. Related policies address worker status and food safety standards, but do not necessarily target worker occupational health and safety directly. These include Good Agricultural Practices (GAP), the Food Safety Modernization Act (FSMA), the H-2A guestworker program, and the Farm Workforce Modernization Act of 2019.

Besides policy efforts, educational initiatives have also developed to reduce adverse health and safety outcomes. Innovations in technology show promise to help address challenges in reaching populations with little available time as well as overcoming language and literacy barriers [12,13]. Some efforts have focused on developing health care providers’ ability to identify and respond to the occupational stressors of agricultural workers in addition to training farmworkers [14,15,16]. New models of the process of influencing health behaviors attempt to account for the factors that contribute to risky behavior in an attempt to better understand these elements in conjunction with the impacts of educational programs [17].

Despite efforts to address the high levels of occupation-related illness and injury through regulatory and educational efforts as well as training programs, these issues continue to exact an elevated cost to human health and economic productivity worldwide. These outcomes have been worsened by the COVID-19 pandemic, as many of the Center for Disease Control (CDC) guidelines have been difficult to implement among agricultural workers. Based on the number of factors intersecting to contribute to negative health outcomes of agricultural professionals, we sought to better understand the limitations to previous and current attempts to improve agricultural health and safety, by gathering the perspectives of agricultural professionals. Specifically, we sought to identify the perceived barriers and enablers to achieving occupational health and safety in agriculture in the southeastern U.S. coastal states. We anticipate that our findings will have broader significance in the international context as many challenges to achieving agricultural occupational health and safety extend globally [1]. Due to the timing of our interviews, we also asked our respondents about the response to the COVID-19 pandemic. Although this was not the initial nor primary focus of our research, we adapted to the circumstances and included questions related to the pandemic as well, particularly in light of its relevance to health and safety.

Therefore, the present study aimed at answering the following four questions: 1) How do farm and forestry stakeholders located in the southeastern United States perceive the agricultural industry from an occupational health and safety standpoint? 2) How do they think their industry is trying to improve occupational health and safety? 3) What are the main barriers to implementing health and safety in their industry? 4) What measures do they think could be taken to ensure better health and safety for people working in their industry?

## 2. Materials and Methods

This research is part of a larger study conducted by the Southeastern Coastal Center for Agricultural Health and Safety funded by the CDC and National Institute for Occupational Health and Safety (NIOSH). We used a qualitative semi-structured interview approach, which was deemed to be the most appropriate, considering its flexibility in allowing researchers to deepen their conversation with their participants, when necessary. Bernard posited that “If you are trying to understand a behavioral process, then focus on qualitative data” [18] (p. 604). We selected the participants (N = 11) based on purposive and snowball sampling techniques but with two *a priori* inclusion criteria. These criteria required that all the participants be involved in the agricultural industry and living in one of the six states covered by the grant funding this study. Such sampling techniques proved to be viable, since qualitative study is not as constraining as quantitative studies in terms of representativeness [19]. Additionally, 11 participants were determined to be adequate per Patton, who suggested that more in-depth information could be obtained with a smaller sample than the broad findings of larger samples [20].

Upon approval of both the pre-testing and the final phases of the interview protocols from the University of Florida’s Institutional Review Board (IRB), we assigned only one member of the research team to conduct the interviews. We did so to ensure consistency in the process and to avoid interpretation related to variations in question delivery, such as voice and tone changes, especially considering interviews were conducted via Zoom, an online platform [21]. We collected the data over a month-long period, namely August 2020. Based on our inclusion criteria, we purposefully selected five participants from the advisory board of a research and education center dedicated to addressing the occupational health and safety needs of people working in agriculture in the southeastern coastal states. Then, we used snowball sampling to recruit the remaining participants, still following the same inclusion criteria. Overall, the participants included extension agents, farmworkers’ advocates, growers, farm labor contractors, and representatives from the worker safety program, and the forestry and fruit and vegetable associations. They represent both the farm (n = 7) and the forestry industries (n = 4). They were all college educated and most of them were over 50 years of age. Their strong involvement in, long experience in, and great knowledge of the industry shed light on the multiple questions that this study addresses.

We conducted separate, semi-structured individual interviews except for one case where two colleagues decided to be interviewed together. This approach, thanks to the use of open-ended questions [18] and its flexibility, allowed us to probe as needed for clarification throughout the conversation [20]. Thus, that limited the potential to impose our own perceptions of the meaning of the data, which could have compromised data validity [22]. We conducted most of the interviews at night based on the participants’ availability. With the participants’ verbal consent, and thanks to Zoom’s built-in recording and transcription features, we both recorded and transcribed all the interviews. The latter took the form of guided conversations, which lasted 45 to 90 min.

The interview protocol included eight questions related to occupational health and safety, including follow-ups pertaining to COVID-19, as we collected the data in the midst of the pandemic. It was divided into three main sections (see Appendix A for the instrument). The first section had two items and addressed farmers’ and foresters’ general descriptions of the industry. The second section had a set of five questions, and covered barriers and the enabling conditions related to agricultural health and safety. The final section asked the participants for suggestions and thoughts that they assumed worth addressing based on the research protocol.

Acknowledging that qualitative research is prone to criticism of being undisciplined and prone to selectivity bias [21] because of the absence of a universally accepted analytic routine [23], the team ensured a sound data management system from the outset, which followed the five analytical stages recommended by Yin [23]. We analyzed the data using the constant comparative method [24,25]. First, the interviewer started with data cleaning, which consisted of revising each interview’s transcript generated by Zoom while watching the recordings to ensure accuracy in the verbatim transcriptions. We combined all the interview transcripts into one dataset that was shared with the research team for familiarization and data cleaning. Second, guided by the research questions, two team members conducted parallel coding to increase internal validity [20,26,27] and to cast a wider analytic net and provide a “crowd-sourcing reality check” for each other according to Harding, as reported by Irani et al. [28] (p. 72). Using color coding and data constant comparisons, each coder developed their initial codes with eventual supporting quotes to enhance the efficiency of the process once the coding process was finalized. During that phase, the coders met to compare and discuss their initial codes and to agree on adopted codes. We purposefully used adopted codes versus final codes because no code is ever final since different people may code the data differently and come up with new codes [29]. Third, we proceeded to the grouping of similar codes into themes [26,30]. Fourth, we revised and refined the themes as necessary. Fifth, we defined and named the themes before submitting them to the team leader for feedback. It is important to mention that the coders sought the team leader’s feedback throughout the stages of the process. Additionally, the research team created an audit trail using Microsoft Excel, group emails, and Microsoft Teams, an online collaboration platform.

We endeavored to establish trustworthiness throughout the entire process to enhance the credibility of the findings [31]. Beginning with the quality of the interview questions themselves, we tested them to ensure that they were not leading questions that might solicit a desired, but not necessarily accurate, response [21]. As discussed above, we recorded all the Zoom conversations to ensure accuracy in the data description, as a way not to compromise the data transcription. Additionally, the use of open-ended questions coupled with follow-up questions helped us overcome any eventual interpretation validity [21]. Acknowledging the impact of the researchers’ inherent bias on data interpretation, once the data were collected and transcribed, we shared the data manuscript (almost 200 pages) among all three team members for data familiarization, and testing for the flow in the participants’ responses or any eventual missing data. Moreover, we checked for meaning clarification when going through the dataset [22] to avoid imposing our own meaning to participants. Fortunately, we did not face any meaning clarification situations.

## 3. Results

Upon analyzing the participants’ responses, the findings were grouped into three main themes: description of the agricultural industry, barriers, and enablers. These include the description of the current state of occupational health and safety within the industry, the barriers faced by the industry to improving occupational health and safety, and the enablers or measures that are being taken, or could be taken, to ameliorate the adverse occupational health and safety outcomes in the industry. All three groupings include components related to COVID-19′s impact.

### 3.1. Description of the Agricultural Industry

Participants’ description of the industry in terms of occupational health and safety was coded into three main subthemes: regulations, inherent danger, and attitudes and education. These subthemes were found to be interdependent, meaning that minimal regulation of the industry, farmworkers’ level of education, and farmers’ attitudes increased participants’ perceptions as to the level of risk inherent in the industry. In addition to these subthemes, participants also reported on other contextual factors, such as the dependence of the global food supply chains on immigrant (often undocumented) workers, and the influence of farm diversity on occupational health and safety. It is noteworthy that participants did not necessarily frame these factors as positive or negative with respect to occupational health and safety, but they emerged through analysis of their responses as relevant. In this sense, farm diversity encompasses farm size and operations, which includes management, equipment, and crop types. In terms of farm management, the newer generation of farm owners tends to be more educated than the predominantly older generation currently leading the agricultural industry. Findings revealed that most of the farm business successors are at least college educated and are more willing to embrace health and safety practices. Diversity in terms of equipment refers to the various tools and machinery required in different types of operations. The type of equipment used impacts the occupational safety of farm workers. For example, in addition to the inherent dangers of operating heavy machinery, workers running such equipment tend to be more prone to skip water breaks. With respect to diversity of crop types, participants noted that this factor was especially relevant in responding to COVID-19. For example, participants said that it was more difficult to maintain social distancing in row crop operations because of planting distance.

#### 3.1.1. Regulations

Although agriculture is one of the most dangerous industries, it is, particularly in Florida, primarily regulated through the U.S. Department of Agriculture (USDA) and the Florida Department of Agriculture and Consumer Services (FDACS) with limited intervention from OSHA, which tends to focus more on the construction industry, as one participant said. Therefore, the latter may only intervene in cases of serious or grave accidents. That can make it even more challenging for at-risk agricultural stakeholders including farmworkers, farmers, and even contractors or intermediaries to maintain standards for occupational health and safety. One participant shared, “there’s not enough compliance with pesticide regulations. There’s not enough enforcement of pesticide regulations.” However, the new Agricultural Worker Protection Standard (WPS) about health and safety may provide more protection for farmworkers.

#### 3.1.2. Inherent Danger

Agriculture is typified as one of the most dangerous industries for workers because of its inherent level of risk and workers’ lack of access to health care. Indeed, farmworkers are prone to high-risk diseases, due to exposure to the effects of heat and pesticides. Participants revealed that the most two common diseases faced by farmworkers are acute skin diseases and heat-related illnesses (HRI). The following expert gives us an idea of the disease risk faced by those involved in agriculture:


*“There are lots of acute effects. Almost every farmworker that works in farm work at some point in their life has rashes, that are related to pesticide exposure, of which acute skin diseases and heat related illnesses are the two common diseases.”*


Participants continued to say that symptoms related to exposure to extreme heat or pesticides may vary from the simple to the more complicated. Simple symptoms may include headache, nausea, dizziness, and vomiting. Those resulting in long-term consequences may include infertility, spontaneous abortion, miscarriage, autoimmune diseases, kidney disease, Parkinson’s disease, and even cancer. One participant reported that while they were conducting focus groups with farmworkers, a woman stated:


*“Oh, well, you know, we wear a hat and we wear long sleeves and we try and take our clothes off before we touch our kids and then one woman in the focus group said. ‘But we still have to breathe the air.’”*


Farmworkers often work in very hot temperatures, exposing them for long hours to the effects of heat and solar ultraviolet radiation. Therefore, it is required that farmers provide shaded tents for them to rest and take water breaks. Participants noted, however, that sometimes workers do not bother to pause for water, especially those paid by piece rate and those working on machinery. Workers paid based on their productivity tend to forgo breaks, pushing themselves more and more in order to make as much money as possible while neglecting their own health. In some cases, operators of heavy equipment are not allowed to bring water on board with them. Therefore, they may choose to bypass their water break, when considering the time it takes to stop and pause for water, leading them to work for the entire day without water. As a result, they might become dehydrated, which is a vector for a wide range of acute and chronic disease conditions.

Although farmworkers are exposed to high-risk diseases, they may not have proper access to health care and health insurance. When they do, they do not have much coverage. Per the National Agricultural Workers Survey (NAWS), only 47% of workers reported having health insurance, with only 43% reporting that they received government-provided health insurance and 29% employer-provided health insurance [32]. The participants reported that lack of legal documentation is a determining factor in whether this category of workers may access health care and health insurance. Therefore, besides going to work, they limit their interactions with organizations requiring identity documentation, including accessing health care, even if they might be eligible for some services classified as humanitarian help. They live in fear of being trapped by immigration and being deported. Additionally, they are afraid of using free available services because of the public charge rule, which penalizes ineligible people for using public health care services. A participant shared that:


*“Some workers know where the clinic is, they go to their clinic. Right now, the current administration [as of 2019] and the federal government have caused this thing called the public charge rule. Right now, a lot of farm workers are afraid to even go to the clinic because they’re afraid they’re going to be penalized because they’re using healthcare services because of the public charge. So, sometimes people are afraid to go to the clinic because they’re afraid of immigration. And sometimes they’re afraid to go to the clinic, they don’t have health insurance. And sometimes they’re afraid to go to the clinic because there has been rumor that the clinics are going to hurt them.”*


Among documented farmworkers, including the H-2A workers, many of them do not have health coverage, because health insurance is perceived as being too expensive. For example, the H2A workers, through their recruitment contract, are legally eligible for health insurance, but they must pay their contribution. However, due to the high cost, most of the time they do not sign up. Furthermore, many immigrant workers are familiar with free public health insurance systems available in their home countries, even though these are sometimes inadequate. Two different participants reported that:


*“The H2A workers and some other labor force are now entering legally the United States. They got different choices for insurance. Some companies provide insurance. So, they get different choices. Some of them, because they don’t see the necessity, they don’t buy it. Health insurance is the last thing that workers want to spend money on because they are not sick if you are not sick, why you’re going to spend money on health insurance. It’s like paying an insurance for nothing. People don’t see the necessity because many of these countries get the healthcare things taking care of for people, if you want to call it that way.”*



*“So, they don’t see the necessity they don’t buy it. They wait the last minute to make that decision to make that phone call, because they know it’s going to cost a lot, you know. So, that’s the problem. They don’t see the necessity of having primary care. Also, sometimes it’s not the best; the rural community doesn’t have the best, you know. You got the migrants’ clinic, but the network is not the best either.”*


#### 3.1.3. Attitudes and Education

Our findings suggested that farm owners’ attitudes toward health and safety often dictate their farmworkers’ attitudes. Additionally, farmworkers’ motives, experience, and culture, including their education, attitudes, beliefs, and stigmas are determinants for their behavior toward health and safety. Participants were unanimous about their perception that youth were more risk tolerant than older people, even amid the COVID-19 pandemic, regardless of their sex. However, different motives drove young men and young women as to levels of acceptable risk. Young males’ level of risk was linked to their level of energy and their desire to make money. A participant, talking about farm workers’ break time, said that: “Remember, these are young 20 something year old guys that are extremely in good shape.” The female participants, on the other hand, were cognizant of their lack of education. One participant stated that:


*“Okay, so we did a focus group once about eight years ago, with farmworker women found with several focus groups. And what we found is that younger farmworker women had less knowledge and less concerns about or fears of or worries about pesticide exposure.”*


Additionally, new H2-A workers and younger workers took more risks and were less cautious as well, because of their lack of knowledge of the tasks they would be undertaking. Overall, participants reported men to be less concerned about health and safety than women. Moreover, as many farmworkers originate from countries predominated by communalism, mainly from Latin countries, their culture of living in close proximity with family members tends to impact their compliance with CDC guidelines during the pandemic, particularly with social distancing.

#### 3.1.4. Summary: Description of the Agricultural Industry

Participants shared information regarding the state of the agricultural industry in general, providing a context to understand the barriers and enablers to advancing occupational health and safety within the profession. This context includes the regulatory environment, intrinsic dangers of the current industry standard practices, including exposure to heat, pesticides, and low levels of health care access, and prevailing attitudes towards risk among segments of the agricultural worker population. The wide range of diverse agricultural operations also emerged as an important contextual factor.

### 3.2. Barriers

Based on our analysis, we found that the barriers to implementing occupational health and safety practices in agriculture are varied. There is no consensus about the prioritization of barriers. One participant summarized the situation, “there’s so many factors, there is not a single or 1, 2, 3 priority factors.” The main subthemes of barriers that emerged from analyzing the interview transcripts are education, lack of access to health care, logistics, discrimination, cultural competency, economic considerations of farm owners and workers, and transparency of labor contracts.

#### 3.2.1. Education

Our findings indicate that there are many impediments to the education and training of farm workers, which poses a challenge to the effective implementation of health and safety practices. These impediments include the transience of farm workers following the agricultural production and harvesting cycles according to the seasons, the demanding nature of agricultural work (leaving little time and energy to pursue further education or training), language and literacy barriers, and a lack of accessible resources according to the technological literacy levels of workers. All these factors together create significant barriers to developing effective educational programming. One participant commented,


*“They come to a seasonal work and they move to another state, then come back again…. So, that also can affect how you can design a program to ensure that there is a continuing education program that would increase their … opportunities to improve their life.”*


#### 3.2.2. Health Care Access

In terms of health care, participants indicated that access is limited by legal and economic barriers. This is a serious impediment to the general health and safety of agricultural workers in the U.S. Some migrant workers are undocumented and therefore face significant risks in seeking medical care. As noted by one participant, “they might not be feeling very comfortable to show up in a clinic because they may be undocumented.” Other workers, even with documentation, such as those working under the H2A system, may decline to purchase health insurance due to the cost. Health care access was also found to vary between workers depending on their position in the enterprise, as noted by a participant,


*“I think there is probably less health insurance in agriculture at the lower levels of employment than there is in other industries, simply because the farmers can’t afford to provide health insurance to all their workers ... because of the nature of agriculture and the earnings in agriculture.”*


Related to this point, participants shared that the cost of seeking medical attention is often spread beyond the patients themselves, since, due to limited transportation options, another worker may be required to assist:


*“They don’t have transportation, you know. … somebody has to miss a day of work with them to bring them to a clinic.” These circumstances contribute to a lack of preventive care for agricultural workers. “The farm workers go to the doctor when they are dying. So, they don’t do preventive care too much. You know, … ‘oh, I’m okay. I get a little pain here, and don’t want to call 911 until the last minute.’”*


This example also demonstrates a logistical challenge of transportation for farmworkers.

#### 3.2.3. Logistics

We found that logistics pose barriers to accessing educational opportunities, health care, and the occupational health and safety information for agriculture workers in general. In addition to available transportation, another logistical problem is inadequate housing, including lack of access to reliable internet. Additionally, workers tend to have more occupants per dwelling and often dwell in substandard conditions.


*“Well, the transportation is being an issue now with COVID-19. We have been putting people inside the bus... So, that’s a problem for the COVID-19... So, it’s all linked, for housing... you also have to see the other ones who are renting. They want the rent to be lower. So, they get together and share the rent. So, they get in a trailer or lower because they are sharing even though they will never inform the landlord how many people are living there. So, they are there, if one of them get sick you don’t have the easy way to isolate at first.”*


This quote highlights the underlying role logistics as a barrier faced by agriculture sector workers and their employers, and in the face of COVID-19, how difficult it became for workers to follow the recommended CDC guidelines such as social distancing and proper sanitation methods.

#### 3.2.4. Discrimination and Cultural Competency

Another issue that emerged in our findings involved the connected elements of discrimination and cultural competency. Participants shared examples of classism, racism, stigma related to cultural norms, and misunderstandings related to a lack of cross-cultural awareness. One participant remarked that,


*“There is also an issue of classism. We were in one nursery where the supervisors were the same ethnic group. They demeaned the workers under them because they were less educated. There were really big class issues among some of the workers and the supervisors, because the supervisors look down on the workers and call them bad names. There’s racist issues, very big problem even between growers and workers, but also among workers sometimes.”*


Issues of discrimination may foment a dismissive attitude towards worker health and safety on the part of farm supervisors. The same participant commented,


*“The grower put on the EPA video because it’s required under the new worker protection standard. The students were going to work for only four hours, the grower had to show them the EPA video while the students were there and he said, ‘Well, you know, I have to show you this video, but you know it’s really nothing to worry about… you can drink water and get… chemicals, too.’ So, it’s not a big deal. So that was the attitude of the grower!”*


Another culturally relevant finding is the perception on the part of several participants that the workers themselves fail to take advantage of protections available to them due to their susceptibility to stereotypes and stigma. Specifically, the element of *machismo* presents a barrier to effective occupational health and safety. A participant noted,


*“I was talking with a Hispanic farm labor contractor recently. And his comment was, yeah, it’s the macho attitude is a bit of a barrier. It’s a struggle. They don’t want to be seen wearing a mask. They see it as being weak if they’re wearing a mask. So, so that may be that may be an issue. Yeah. Again, that’s the cultural aspect of it.”*


#### 3.2.5. Economic Considerations

In the discussion of barriers, economic considerations on the side of the farm workers as well as that of the farm owners further impede the effectiveness of health and safety efforts. On the part of the farm workers, as touched on above, there is an intense financial pressure to support themselves as well as family members living with them and, in many cases, in their home countries. This often means workers will push themselves past physical limits—working long hours, skipping water and shade breaks, working even when sick—to maximize their income. This is particularly true for the many workers who are paid by the piece (the quantity harvested) as opposed to an hourly wage. Participants observed that “you’re talking about a very disadvantaged community, low income, poverty levels, lack of transportation,” and


*“These guys, you know, if they get a cold, they don’t go to the doctor, they want to work, they came here to make money. They make really good money when they’re here, certainly, in comparison to what they are able to make where they came from.”*


These issues have been highlighted by the COVID-19 pandemic. As noted by Irani et al. [28] (pp. 77–78), “workers’ financial insecurity sometimes pushed them to prioritize work over their health. They might be reluctant to get tested for COVID for two main reasons: (1) they did not want to know their status, because a positive test meant a minimum of 14 days out of work while they got paid by the piece and (2) they did not have health insurance to cover the test.” In the case of the farm owners, many operate on narrow margins, allowing for very little additional expenditure, particularly in the case of upfront investments for health and safety, which do not necessarily directly translate to additional revenues. A participant shared the following regarding the perspective of farm management, “Because this [purchasing personal protective equipment] will be seen as impacting their bottom line. We will be seen as costing them time, costing them money.” Therefore, in this sense, both the farm owners and workers might be motivated to sacrifice occupational health and safety to alleviate economic tensions.

#### 3.2.6. The Labor Contracting System

Finally, the labor contracting system in place contributes to a level of confusion as to where responsibility lies for providing health and safety measures to agricultural workers. In some cases, the farm owners defer responsibility to labor contractors, who are technically the employers. In others, the labor contractors defer to the farm owners and managers, who they consider to be the responsible party. This situation is captured by a participant, “Lot of times the grower … doesn’t even know what’s going on. Sometimes they’re being pushed by the labor contractor.” Another participant explains,


*“A large majority of the workforce, particularly in Florida is managed by farm labor contractors. Nationally, farm labor contractors are responsible for 41% of the H2A workers that come into this country. That’s about half the workforce. In Florida, I suspect it’s a higher percentage just because we are more heavily dependent on the farm labor contractors in the state. So, you know, it appears to me that the answer to that is to get to that workforce through the farm labor contractors. They will listen to those bosses. They will respond to those people, the farm labor contractors whom they are working for and who they, quite frankly, feel a little closer to. So, it seems to me that is sort of the key to getting to the workers themselves is through the farm labor contractors, the people they work with.”*


#### 3.2.7. Summary: Barriers

In summary, there are many interconnected and overlapping barriers present in the agricultural sector that may influence the occupational health and safety of industry workers. It is difficult to determine a clear-cut order of importance of these barriers. The primary subthemes of barriers identified include lack of access to education, health care, adequate housing and transportation, discrimination and stigma related to cultural stereotypes, economic challenges of both farm workers and owners, and a lack of accountability to provide adequate health and safety efforts. While these barriers create a complex and challenging situation to navigate, findings also included several promising factors to facilitate and enable positive change and increase occupational health and safety in the agriculture industry.

### 3.3. Enablers

The analysis produced three subthemes of enabling factors for successful integration of occupational health and safety measures into the agricultural industry related to education, regulation, and preventive health care. These include opportunities to improve education through increased and more effective outreach and emphasizing the connection between health and economic opportunity. The role of regulations emerged as an important element in incentivizing and enforcing worker health and safety practices and standards. Improving access to health care, especially preventive care, also emerged as an opportunity to advance occupational health and safety of agriculture workers.

#### 3.3.1. Education

Findings suggest that education constitutes a potential and sound way to improve health and safety in the agricultural industry. This has been reinforced by the COVID-19 experience. Although COVID-19 has created a lot of disturbance in the industry, as in every business sector, participants agreed that it provides an unprecedented opportunity for training providers and extension specialists to revamp their programs and adapt their approach. Participants echoed that COVID-19 facilitated outreach and communication by making virtual training a standard practice as opposed to before the pandemic. One participant stated:


*“We’ve moved our longer training program to an online format for this year. And I suspect, and talking to folks, as people get more and more comfortable, that online virtual meeting will become more of a staple even after we go back into space where we can get together.”*


Virtual training sessions were revealed to yield higher efficiency through an increased number of participants. Agricultural stakeholder participants, including farmers, did not question the use of online training. They seemed to be comfortable with such training settings. Therefore, some participants expect big companies to end up using technology for distance learning, marketing, and sales. Another participant reported that:


*“We are in the process to start putting a class together. We’re going to have a class on the 19th and the other one on the 25th. We got the Zoom meeting for the 19th. I think we have to open a second for the 25th, because there are so many people already. We are close to 200 farmers already. We’re talking about supervisors. We are not talking about farmworkers. We’re talking about people in charge, interested to get that training.”*


As highlighted in the section on barriers, participants mentioned the importance of topics related to racial and ethnic discrimination, classism, and inadequate sanitation, which constituted impediments to proper implementation of occupational health and safety, including the new CDC COVID-19 guidelines. Therefore, participants suggested that developing educational materials in these subject areas would help to fully address occupational health and safety in the agricultural industry. Furthermore, participants recommended holding training sessions and providing training material to people in their own languages to ensure optimal knowledge transfer. Two participants added:


*“Some of the companies provide training, but they never verify that the people understand the training. … They put a video and among the workers, you’ve got different language. But, we got issues with Creole while you see more Creole people, working in the industry. I got surprised. Two years ago, when I went to the farm. I never saw so many people with Creole and there is a bunch of people with Creole and they need education.”*



*“We see workers who don’t speak English. So, providing training opportunities for them in other languages outside of English would be ideal.”*


#### 3.3.2. Regulations

Several participants focused on the role of regulations in improving worker health and safety. This subtheme contained several components of regulation, including improving the application and enforcement of existing standards, such as the improved WPS, GAP, and FSMA, as well as incorporating stronger protections for farm workers. One participant proposed,


*“I think we need policies that protect workers. Workers are being affected by poor health, poor housing, crowded housing, by lack of access to educational material, by heat. We need to come up with protection plan for workers. Oftentimes I feel like a lot of the laws are designed to protect growers from lawsuits but I think in doing that, we oftentimes also ask again why these protections are in place.”*


Additionally, broader aspects of regulations were addressed, including a call to acknowledge agricultural workers as a skilled labor force, to standardize a process of on-farm reviews, and to address system-wide issues that inhibit the ability of the industry, as a whole, to provide adequate health and safety measures. This sentiment is captured by the following statement,


*“Well, we need to have a different culture and our entire agricultural system. This culture of production… I mean, part of the problem is that the growers have to have a profit margin and they have inputs of fertilizers and pesticides and machineries. And statistics show that economically the only place that they can cut costs when prices go down so that they can make a profit, the only place they can cut costs is supposedly is labor. The costs on the paying of the workers and also in the health and safety of the workers. So, there’s something wrong with our system. If growers are not being able to make enough money …so, it’s all along the food chain.”*


#### 3.3.3. Healthcare and Prevention

Healthcare access emerged as a critical element of improving agricultural workers’ health and safety. Participants conveyed that addressing agricultural occupational health and safety requires that farmworkers’ healthcare be a priority. Some of the opportunities to improve in this area include instituting positions in farm operations to oversee and manage health and safety training and implementation, providing preventive care for uninsured and underinsured workers, and working with healthcare providers to address the specific risks of agricultural work. For many farms, health and safety training may be a one-time event rolled into the overall operation. Respondents shared in interviews that the creation of a health and safety manager position may help to bring more focus and consistent implementation of best practices,


*“These days you need a position for a safety and health manager and that person will oversee the safety and health program for the company. When you see company, they don’t have that person in place you will see more problems, you know.”*


One of the opportunities for a health and safety manager is to provide an increased focus on preventive care for workers. This is critical for the uninsured and underinsured agriculture workers discussed in the barriers section above. An emphasis on preventive healthcare reduces the need for emergency services. There is an opportunity to expand this focus, especially given the economic incentives of farm employees to work as much as possible. One participant shared a positive example of working with not-for-profit volunteer healthcare providers to provide preventive care for their workers,


*“We participate with a group of doctors that are better volunteers that come to our job site and test workers and provide dental care once a year, and that’s I think that’s a very much a positive thing. But if that could be somehow provided to these workers that don’t have health insurance, it would be a great benefit to everyone.... One time a year, we would bring that volunteer group in and they would provide medical services, shots, dental care, and all kinds of things for them.”*


This example represents a possible way forward in reaching more agricultural workers where they are in terms of addressing their medical needs in their economic reality. Finally, in addition to working with healthcare professionals to provide more accessible services, another opportunity is in working to educate healthcare providers about the specific risks and dangers of agricultural work. For example, the common illnesses associated with pesticide exposure may be misidentified as a common rash. Incorporating knowledge of the industry’s practices and associated risks into the training of nurses and doctors, especially those serving in clinics in rural and agricultural areas, would be advantageous for the health of agricultural workers. As relayed by one participant,


*“The clinics ...are not trained in occupational and environmental health. ... Their intake forms… ask you about your mother’s medical history, your father’s medical history, your medical history, they don’t ask you where you work. ...We did a training once in Miami. Some doctors that have clinic that the farm workers go to in Miami. We did an hour and a half long training at the end of the training, one of the doctors came up to me and said, you know, I had a farm worker patient about two weeks ago that had rash all over his body and I just diagnosed it as acute dermatitis gave him some cream and sent him home. But after hearing you talk, I think now that was related to pesticide exposure and I should have put it down as pesticide exposure, not acute dermatitis, and I should have reported it and I should have told him to take protections against pesticide.”*


#### 3.3.4. Summary: Enablers

Overall, participants reported that education remains a viable path forward to increase occupational health and safety in agriculture, particularly with respect to the potential for virtual engagement, as highlighted by the COVID-19 pandemic, and the need to develop programming to address discrimination by class, race, and ethnicity. Similarly, regulation represents an opportunity to systematically improve the processes governing health and safety practices. Finally, there is potential to enhance the role of preventive healthcare and healthcare access for workers through partnerships with healthcare providers, including agriculture-specific medical indicators such as rashes due to pesticide exposure.

### 3.4. Results Summary

To summarize, there were three themes to the findings: the description of the current state of occupational health and safety within the industry, the barriers faced by the industry, and the enablers or measures that are being taken or could be taken to ameliorate the adverse occupational health and safety outcomes in the industry. Each of these themes contained subthemes. The subthemes for the industry description included regulations, inherent danger, and attitudes and education. The subthemes for barriers contained education, lack of access to healthcare, logistics, discrimination, cultural competency, economic considerations of farm owners and workers, and transparency of labor contracts. The subthemes for enablers were education, regulation, and preventive healthcare.

## 4. Discussion

The present study sought to better understand the limitations to previous and current interventions to improve agricultural health and safety from the perspectives of agricultural stakeholders. More specifically, this study aimed at identifying the perceived barriers and enablers to enhancing occupational health and safety in agriculture, notably in the southeastern U.S. coastal states. Four questions helped meet this objective. First of all, we asked the participants about how farm and forestry stakeholders, located in the southeastern United States, perceived the agricultural industry from an occupational health and safety standpoint. Second, we discussed with them their perception of their industry’s (farm, fishery, forestry) efforts to improve occupational health and safety. Third, we covered the main barriers to implementing health and safety in their industry. Finally, we collected their suggestions about what measures they thought could be taken to ensure better health and safety for people working in their industry.

Upon analyzing the participants’ responses, three main themes stood out. They include the description of the current state of occupational health and safety within the industry, the barriers faced by the industry, and the enablers or measures that are being taken or could be taken to ameliorate the adverse occupational health and safety outcomes in the industry. As this study was conducted in the middle of the COVID-19 pandemic, and acknowledging the drastic change brought by such a plague in people’s lives, we made sure to weigh in before and during COVID-19 during each team analysis.

Consistent with existing literature, the study findings to describe the agricultural industry were categorized into three main subthemes. These include regulation, inherent danger, and attitudes and education, which are all interdependent. They conveyed that the lack of regulation of the industry, farmworkers’ level of education and attitude bundled together to define the danger inherent in the industry. This finding correlates with existing literature, which identifies agriculture as the second most dangerous industry after construction [3]. Agricultural workers are exposed to a high-risk environment. Exposure to pesticides causes all types of diseases, from skin to respiratory and cardiovascular problems [6]. Additionally, exposure to warm temperature provokes heat-related illness that can be deadly [4]. When factoring in the COVID-19 impact, participants considered COVID-19 as an exacerbating risk factor of the industry.

The findings revealed many interconnected and overlapping barriers present in the agricultural sector that may influence the implementation of health and safety. The participants expressed five most prevalent inhibitors to complying with occupational health and safety: education, healthcare access, logistics, discrimination, as well as cultural competency, economic considerations, and the labor contracting system. The farmworkers’ 8th grade average schooling level [32], coupled with the demanding nature of agriculture in terms of time, notably 9.2 h a day [33], and fatigue, is not conducive to complying with occupational health and safety measures. Additionally, lack of healthcare access, due to legal and economic barriers, constitutes bottlenecks to occupational health and safety implementation. This speaks to Guild and Figueroa, who reported that undocumented immigrants tend to limit their errands for fear of encountering immigration enforcement because of the increased immigration enforcement [34]. Additionally, farmworkers’ responsibility toward their own and extended family members pushes them to overextend themselves to make more money at the expense of physical health. Moreover, logistical challenges, including housing and transportation, are a big piece of the puzzle, intersecting with other factors. For example, available transportation might facilitate healthcare access and continued education. Additionally, living in reasonable quarters with access to the internet might allow them to attend online English courses. Furthermore, participants revealed the manifestation of classism, racism, and stigma related to cultural norms and misunderstandings related to a lack of cross-cultural awareness when in the field. Finally, the labor contracting system in place contributes to a level of confusion as to where responsibility lies for providing health and safety measures to agricultural workers. This resonates with the NAWS reports, which highlighted that only 47% of workers reported having health insurance [32]. Among them, 43% reported that they received government-provided health insurance, 29% received employer-provided health insurance, and the others from other sources such as through insured spouses.

Although the barriers to implementing occupational health and safety seem overwhelming, participants identified key factors to overcome them and set the stage for successful integration of health and safety in the industry. The first factor relates to the opportunity to improve health and safety educational efforts by expanding the types of outreach, covering critical subject areas, and reaching audiences in appropriate languages. The second factor addresses the opportunity to increase worker health and safety through regulatory measures, including enhancing existing standards and expanding the scope of regulatory action to reconsider agriculture workers as a skilled labor force and examine the system along the entire food chain. Finally, there is significant opportunity for improvement through improving access to preventive healthcare and education of healthcare providers on occupational hazards common to agricultural work.

A limitation of this study is that our sample did not include any farmworkers. Although some of our interviewees represent farmworker advocacy groups and others who work closely with farmworkers, including the perspectives of farmworkers directly would add an important perspective of understanding to the topic. Additionally, although we sought to address COVID-19 directly in our interviews, it is difficult to determine the impact of the ongoing pandemic through the findings reported. Therefore, further assessments are necessary to observe shifting perceptions and attitudes as they continue to evolve. Finally, the deep and rich discussion of many of the barriers and enablers to occupational health and safety in agriculture produced through this research would benefit from a comprehensive quantitative study. Our understanding would also benefit from future research to develop a questionnaire based on our findings to be used with a larger sample of agricultural stakeholders.

## 5. Conclusions

This qualitative inquiry provided insight into the barriers and enablers of occupational health and safety of agricultural workers in the southeastern U.S. coastal states, which may be analogous to international experiences. Findings revealed the perceived barriers of education, lack of access to healthcare, logistics, discrimination, cultural competency, economic considerations of farm owners and workers, and transparency of labor contracts. Participants reported these barriers to be interconnected and overlapping. Perceived enablers of occupational health and safety included factors associated with education, regulation, and preventive healthcare. We recommend that future research incorporate farmworkers directly and recruit a larger, representative sample of agricultural stakeholders for complementary quantitative analysis.

## Data Availability

The data presented in this study are available on request from the corresponding author. The data are not publicly available to protect the privacy of participants.

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
