# Peer review of "Agricultural Stakeholders’ Perceptions of Occupational Health and Safety in the Southeastern U.S. Coastal States"

_ijerph, 2021, doi:10.3390/ijerph18126605_

Round 1

Reviewer 1 Report

Agricultural Stakeholders' Perceptions of Occupational Health and Safety in the southern U.S. coastal states

The article describes an important issue concerning  occupational health and safety, particularly in the agricultural sector. The region (in the southern U.S. coastal states) in which the data were collected should be mentioned in the title. The regional reference should also be highlighted when citing numbers in the text, e.g., 170,000 fatalities worldwide (line 25) or 573 fatal workplace injuries in 2019 in the U.S. (line 27).

For more information on the grant funding the study, the position and skills of the interviewees, and about the authors or the project itself would be helpful. As said, the topic is of great importance, but the elaboration seems like a student project study, it is not very in-depth, which is pointed out many times in the article by the authors themselves. A description of the research team and the research project is missing. Who is "we"? Many claims may be true, but they are not substantiated. Health insurance seems to be a problem not only for agricultural workers, but in the U.S. in general.

Perhaps the article can provide impetus for necessary further research and, more importantly, improvements in occupational health and safety, particularly in the agricultural sector.

Author Response

Please see attached the responses to reviewer 1

Reviewer 2 Report

This paper presents a qualitative research study focusing the perceptions of agricultural professionals on occupational health and safety issues, including barriers and enablers to improve the current situation.

This is a relevant and up-to-date research topic, and in my opinion this paper introduces relevant information with adequate scientific soundness. In spite of the reduced number of interviews performed (n=11), and of the absence of farm workers in the interviewees – which has been acknowledge as a limitation in the discussion section – the results obtained are consistent with the objectives of the study. The methodology is clearly described and is adequate for the research’s objectives.

I have the following comments / improvement suggestions.
-       In the Introduction section your analysis of the current situation covers exclusively data and information from the US. I believe that the majority of the occupational health and safety issues that you focus are common to the agricultural industry in many other countries. In my opinion you should highlight that your research subject is of international concern, and present data/information from other countries. 
-       Still in the scope of the previous comment, you present numbers of fatal work injuries and heat-related fatalities – these data refer to the US, and this should be clearly stated.
-       You refer frequently the ‘H-2A and H-2B visa programs’ and ‘H2-A workers’ and ‘H2A workers’. You should explain what this means, a non-US reader is not familiar with this terminology.
-       In the last paragraph of the Introduction section you introduce four research questions ‘… including some specifics about the impact of the COVID-19 pandemic’ (lines 95-96). I don’t think that this is the adequate way to formulate your research questions – if there are specific questions regarding the pandemic, these questions should be clearly formulated. If there are no specific questions, I believe that the implications of the COVID-19 pandemic are a current issue in occupational health and safety, and therefore these are already included in the four research questions formulated in this study.
-       Your research question 2 refers ‘farm, fishery, forestry’ (line 99), but the fishery industry has not been covered in this qualitative study (line 128, n=7 for farm industry and n=4 for forestry industry).

Author Response

Please
